# Domain Generalization via Heckman-Type Selection Models

**Hyungu Kahng**[1][*]  **Hyungrok Do**[2]   **Judy Zhong**[2]
[1]Korea University    [2]NYU School of Medicine
hgkahng@korea.ac.kr, {hyungrok.do,judy.zhong}@nyulangone.org

## Abstract

The domain generalization (DG) setup considers the problem where models are trained on data sampled from multiple domains and evaluated on test domains unseen during training. In this paper, we formulate DG as a sample selection problem where each domain is sampled from a common underlying population through non-random sampling probabilities that correlate with both the features and the outcome. Under this setting, the fundamental *iid* assumption of the empirical risk minimization (ERM) is violated, so it often performs worse on test domains whose non-random sampling probabilities differ from the domains in the training dataset. We propose a Selection-Guided DG (SGDG) framework to learn the selection probability of each domain and the joint distribution of the outcome and domain selection variables. The proposed SGDG is domain generalizable as it intends to minimize the risk under the population distribution. We theoretically prove that, under certain regular conditions, SGDG can achieve smaller risk than ERM. Furthermore, we present a class of parametric SGDG (HeckmanDG) estimators applicable to continuous, binary, and multinomial outcomes. We also demonstrate its efficacy empirically through simulations and experiments on a set of benchmark datasets comparing with other well-known DG methods.

## 1 Introduction

In statistical learning theory, the standard assumption behind many supervised learning algorithms is that both training and test instances are independently and identically distributed (*iid*) according to the same underlying data distribution (Vapnik, 1991). In other words, most statistical models assume that the training and test data are both random samples chosen randomly from the same population. Unfortunately, this assumption is often violated in real-world applications rendering model performance to deteriorate on out-of-distribution (OOD) test data (Koh et al., 2021). Recently, the Domain Generalization (DG) problem (Blanchard et al., 2011) has gained particular attention, where it is assumed that learning systems have access to training data sampled from multiple domains, and the ultimate goal is to extrapolate to new instances sampled from previously unseen test domains.

In this paper, we consider DG as a non-random sample selection problem. Let $\mathcal{P}_{XY}$ represent the *population data distribution*, and $S^k$ denote a binary random variable indicating whether a subject is selected from the population into domain $k$. In a random sampling process, $P(S_i^k = 1)$ is independent and identically distributed (*iid*). Under a non-random sample selection, the distribution of $(X, Y)$ in domain $k$ is a conditional distribution of $(X, Y)$ given $S^k = 1$, which often does not equal to $\mathcal{P}_{XY}$. Consequently, this leads to distributional shifts across domains: $\mathcal{P}_{XY}^j \neq \mathcal{P}_{XY}^k$ for $k \neq j$. Mathematically, distribution shifts across domains $\mathcal{P}_{XY}^k$ may contain shifts in distributions of $X$ ($\mathcal{P}_X^k$, covariate shift (Bickel et al., 2009)), and in the distributions of $Y$ conditional on $X$ ($\mathcal{P}_{Y|X}^k$, concept shift (Moreno-Torres et al., 2012)). We present a graphical model in Figure 1 to conceptually illustrate the sources of distribution shifts, assuming the existence of latent factors confounding the relationship between $X$, $Y$, and domain ($S^k$). In Figure 1, $C_1$ represents unobserved latent factors that correlate with $X$ and $S^k$, resulting in covariate shift. $C_2$ correlates with $X$, $Y$, and $S$ simultaneously, entailing both covariate and concept shifts. The goal of DG is to estimate the domain generalizable (agnostic) edge $f : \mathcal{X} \rightarrow \mathcal{Y}$ in the presence of the two types of latent confounders.

---

[*]This work was mostly done while the author was with NYU.

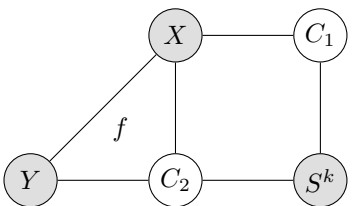

Figure 1: A graphical model illustrating the source of distributional shifts. $X$: covariates, $Y$: outcome, $S^k$: domain. $C_1$ represents latent factors that correlate with $X$ and $S^k$, resulting in covariate shift. $C_2$ correlates with $X$, $Y$, and $S^k$, entailing both covariate and concept shifts. Our goal is to estimate the domain generalizable (agnostic) edge $f : \mathcal{X} \to \mathcal{Y}$ in the presence of the two types of latent confounders.

The vast majority of DG methods are developed to identify $f$ that is robust to $C_1$. However in practice, $C_2$ type of confounders often exist which make $P(S^k = 1)$ related to both $X$ and $Y$. For example, when we train a model to predict tumor status ($Y$) from histological images ($X$) using patients from different hospitals ($S^k$), there may be variations in $X$ due to inconsistent acquisition processes such as staining differences ($C_1$) across hospitals, and differences in patient characteristics such as age, gender, race, and disease severity ($C_2$) that correlate with hospital, covariates and the outcome. As a result, a model trained in an oncology specialist hospital may not be generalizable to a hospital serving veterans. Similarly, when we train a model to predict wealth index ($Y$) from satellite images ($X$) taken from different countries ($S^k$), there may be latent factors such as the economic status ($C_2$) correlating with X, Y and domains simultaneously. Therefore, a model trained on one country may not perform well in another country with a different rural/urban proportion or economic status (Koh et al., 2021).

In this paper, we propose a new class of Selection Guided Domain Generalization (SGDG) models to first estimate the selection probability that an instance is sampled into a training domain, and then use the joint distribution of the outcome $Y$ and selection $S$ to learn a domain generalizable model. In particular, SGDG is built on Heckman's bias correction framework (Heckman, 1979) which is a very powerful tool to learn an unbiased model from non-randomly selected samples in the presence of both $C_1$ and $C_2$ confounders. The unique contributions of this paper are summarized as follows:

- To the best of our knowledge, we are the first paper to formulate the DG problem using a non-random sample selection framework, and to propose a Selection Guided Domain Generalization (SGDG) method under this framework.

- We present a class of parametric SGDG (HeckmanDG) estimators applicable to continuous, binary, and multinomial outcomes[†].

- We demonstrate the efficacy of our method both theoretically and empirically on simulated data and four challenging benchmarks.

## 2 RELATED WORK

**Domain Generalization.** DG has been studied under various contexts. Many studies are devoted to learning *domain-invariant features* which are discriminative and independent of the domain, such as kernel-based methods (Muandet et al., 2013), matching moments (Sun & Saenko, 2016), adversarial learning (Ganin et al., 2016; Deng et al., 2020), entropy regularization (Zhao et al., 2020), and contrastive learning (Motiian et al., 2017; Kim et al., 2021). Other works exploit *invariant causal effects* across domains (Arjovsky et al., 2019; Ahuja et al., 2020; Rosenfeld et al., 2021). Another family of *robust optimization* methods seek to minimize the worst-case error (Sagawa et al., 2020; Xie et al., 2020; Krueger et al., 2021). More recently, other prominent directions of methods improve DG by *model averaging* (Cha et al., 2021; Arpit et al., 2022), gradient matching (Shi et al., 2022), meta learning (Li et al., 2018), data augmentation (Robey et al., 2021), and generating novel domains (Zhou et al., 2020).

**Sample Selection Bias Correction.** Zadrozny (2004) formalized sample selection bias in machine learning terms and presented a bias correction method when selection only depends on the input features. Cortes et al. (2008) proposed a sample reweighting approach to tackle the same problem but assumed the availability of additional data drawn from the true population. Du & Wu (2021) proposed a framework for robust and fair learning under biased sample selection, but assumes conditional independence of $Y$ and $S$ given $X$.

---

[†]code available: https://github.com/hgkahng/domain-generalization-lightning

# 3 SELECTION MODEL-GUIDED DOMAIN GENERALIZATION

## 3.1 DOMAIN GENERALIZATION

Suppose there exists $L$ distinct but relevant *domains* and let $S = (S^1, \cdots, S^L) \sim \mathcal{P}_S$ denote a binary random vector that indicates the domain membership where $S^k = 1$ implies belonging to domain $k$. Let $\mathcal{P}_{XY}$ represent the *population data distribution*, and each domain's data distribution be a conditional distribution of the population distribution given $S^k = 1$, i.e., $\mathcal{P}_{XY}^k = \mathcal{P}_{XY|S^k=1}$.

**Assumption 1** (Mutually Exclusive Domain Membership). *We assume that if $S^k = 1$, then $S^j = 0$ for all $j \neq k$ so that an instance can belong to one and only one domain.*

**Assumption 2** (Independent Domain Sampling Processes). *We assume that $S^k \perp\!\!\!\perp S^j$ for all $j \neq k$.*

In the supervised domain generalization context, we are allowed to observe the joint distribution of $X$ and $Y$, $\mathcal{P}_{XY}^k$ for $K$ out of $L$ domains, and refer the $K$ domains as *source or training domains* observed during the training phase. The remaining $L-K$ domains are referred to as *target or testing domains* whom we may observe in the testing phase. Under this setting, we aim to learn a prediction model that generally performs well on both source and target domains. We formalize the problem as follows.

**Definition 1** (Domain Generalization (Blanchard et al., 2011)). *Domain generalization refers to as the problem of learning $f : \mathcal{X} \to \mathcal{Y}$ that has the minimum expected loss across all possible domains, which can be further summarized as the following optimization problem:*

$$\min_{f \in \mathcal{F}} \sum_{k=1}^{L} \mathbb{E}_{(X,Y) \sim \mathcal{P}_{XY}^k} \Big[ \ell(f(X), Y) \Big] P(S^k = 1), \tag{1}$$

*where $\ell : \mathcal{Y} \times \mathcal{Y} \to \mathbb{R}_+ (= \{r \in \mathbb{R} : r \geq 0\})$ is a loss function and $\mathcal{F}$ is a hypothesis set. We note that some other papers have considered the same problem setting (Muandet et al., 2013; Deshmukh et al., 2019; Blanchard et al., 2021).*

We first introduce a proposition that claims the equivalence of the domain generalization and the risk minimization under the population distribution:

**Proposition 1** (Equivalence of Domain Generalization and Population Risk Minimization). *Problem (1) is equivalent to the risk minimization under the population distribution $\mathcal{P}_{XY}$. That is,*

$$\min_{f \in \mathcal{F}} \sum_{k=1}^{L} \mathbb{E}_{(X,Y) \sim \mathcal{P}_{XY}^k} \Big[ \ell(f(X), Y) \Big] P(S^k = 1) = \min_{f \in \mathcal{F}} \mathbb{E}_{(X,Y) \sim \mathcal{P}_{XY}} \Big[ \ell(f(X), Y) \Big], \tag{2}$$

*which straightforwardly follows from the law of total expectation. We define $f_{\text{PRM}}$ as the minimizer of the population risk minimization problem, as well as the best (hypothetical) model for the domain generalization problem.*

Proposition 1 is important as it establishes that the best model for the domain generalization problem should minimize the risk under the population distribution $\mathcal{P}_{XY}$. However, in practice we have no access to the population data distribution and are only given the source domains during model training. A naive learning model is minimizing the risk under the source domain data distribution:

**Definition 2** (Source Domain Risk Minimization). *Given the source domains whose data distributions are $\mathcal{P}_{XY}^k = \mathcal{P}_{XY|S^k=1}$, for $k = 1, \cdots, K$, we refer to the following learning problem as the source domain risk minimization problem*

$$\min_{f \in \mathcal{F}} \sum_{k=1}^{K} \mathbb{E}_{(X,Y) \sim \mathcal{P}_{XY}^k} \Big[ \ell(f(X), Y) \Big] P(S^k = 1). \tag{3}$$

*We denote its minimizer as $f_{\text{SDRM}}$. The empirical form of $f_{\text{SDRM}}$ is denoted as empirical risk minimization (ERM).*

The generalization performance of $f_{\text{SDRM}}$ depends on how well the source domains represent the population (or the target domains). If the source domains well approximate the population distribution, then the generalization performance of $f_{\text{SDRM}}$ will be sufficiently close to that of $f_{\text{PRM}}$. On the other hand, if $\mathcal{P}_{XY}^k \neq \mathcal{P}_{XY}$, one cannot guarantee that models trained on the source domains to effectively generalize to the population. Therefore, it is necessary to model the selection probability to bridge the gap between $\mathcal{P}_{XY}^k$ and $\mathcal{P}_{XY}$.

## 3.2 SELECTION MODEL-GUIDED DOMAIN GENERALIZATION

Now we derive our selection model-guided domain generalization problem, starting from decomposing the objective function of the domain generalization problem by introducing a *domain selection model* $\boldsymbol{g} = \{g_k : \mathcal{X} \rightarrow [0,1], k = 1, \cdots, L\}$ which predicts the selection probabilities of an instance being observed in domain $k$.

**Assumption 3** (Decomposable Loss Function). *We assume that the loss function $\ell$ can be decomposed into two components - one exclusively about the selection model and the other involving both:*

$$\ell(f(X), Y) = \sum_{k=1}^{L} \mathbb{I}(S^k = 1)\Lambda(f(X), \boldsymbol{g}(X); Y, S^k = 1) + \sum_{k=1}^{L} \mathbb{I}(S^k = 1)\ell_s(\boldsymbol{g}(X), S^k = 1),$$

*where $\ell_s : [0,1]^L \times \{0,1\}^L \rightarrow \mathbb{R}_+$ is the loss function for learning the domain selection models and $\Lambda : (\mathcal{Y} \times [0,1]^L) \times (\mathcal{Y} \times \{0,1\}^L) \rightarrow \mathbb{R}_+$ is the joint loss function, which assigns to the prediction and domain selection models a pair of true outcome $Y$ and domain membership indicator $S$.*

For example, the negative log-likelihood under the probabilistic framework satisfies Assumption 3: if we let $\ell(f(X), Y) = -\log p(Y|X)$, then $\Lambda = -\log p(Y|X, S^k = 1)$ and $\ell_s = -\log p(S^k = 1|X)$, respectively. We provide the full derivation in Appendix A.1 and present a specific parametric form in Section 4.

Under Assumption 3, the population risk can be expanded as follows:

$$\mathbb{E}_{(X,Y)\sim\mathcal{P}_{XY}}[\ell(f(X), Y)] = \sum_{k=1}^{L} \mathbb{E}_{(X,Y)\sim\mathcal{P}_{XY}^k} \left[\Lambda(f(X), \boldsymbol{g}(X); Y, S^k = 1) + \ell_s(\boldsymbol{g}(X), S^k = 1)\right] P(S^k = 1)$$

$$= \sum_{k=1}^{K} \mathbb{E}_{(X,Y)\sim\mathcal{P}_{XY}^k} \left[\Lambda(f(X), \boldsymbol{g}(X); Y, S^k = 1)\right] P(S^k = 1) + \sum_{k=1}^{L} \mathbb{E}_{X\sim\mathcal{P}_X^k} \left[\ell_s(\boldsymbol{g}(X), S^k = 1)\right] P(S^k = 1)$$

$$+ \sum_{k=K+1}^{L} \mathbb{E}_{(X,Y)\sim\mathcal{P}_{XY}^k} \left[\Lambda(f(X), \boldsymbol{g}(X); Y, S^k = 1)\right] P(S^k = 1). \tag{4}$$

Based on the expansion, we introduce our selection model-guided domain generalization problem.

**Definition 3** (Selection Model-Guided Domain Generalization (SGDG)). *Given access to the data distribution of $K$ source domains, $\mathcal{P}_{XY}^k$ for $k = 1, \cdots, K$ and to the unlabeled data distribution from all $L$ domains, $\mathcal{P}_{X|S^k=1}$ for $k = 1, \cdots, L$. We define the selection model-guided domain generalization problem as a joint learning problem of $f : \mathcal{X} \rightarrow \mathcal{Y}$ and $\boldsymbol{g} = \{g_k\}_{k=1}^L$,*

$$\min_{f\in\mathcal{F}, \boldsymbol{g}\in\mathcal{G}} \sum_{k=1}^{K} \mathbb{E}_{(X,Y)\sim\mathcal{P}_{XY}^k} \left[\Lambda(f(X), \boldsymbol{g}(X); Y, S^k = 1)\right] P(S^k = 1) + \sum_{k=1}^{L} \mathbb{E}_{X\sim\mathcal{P}_X^k} \left[\ell_s(\boldsymbol{g}(X), S^k = 1)\right] P(S^k = 1), \tag{5}$$

*where $\mathcal{F}$ and $\mathcal{G}$ are hypothesis sets.*

The SGDG problem is a minimization of the expected loss of the prediction model $f$ under the joint distribution of $X$ and $Y$ of the source domains, and the expected loss of the domain selection model $\boldsymbol{g}$ under the joint distribution of $X$ and $S$. Under this formulation, the domain selection model $\boldsymbol{g}$ guides $f$ to be *corrected* through $\Lambda$, considering the probability of being drawn from certain domains. In Section 4, we will introduce specific forms of this problem under parametric assumptions.

**Theorem 1** (Performance Improvement of SGDG over SDRM). *Let $f_{\text{SDRM}}$ and $f_{\text{SGDG}}$ be defined as in Definition 2 and 3. $f_{\text{SGDG}}$ has lower risk than that of $f_{\text{SDRM}}$. That is,*

$$\mathbb{E}_{(X,Y)\sim\mathcal{P}_{XY}}[\ell(f_{\text{SGDG}}(X), Y)] \leq \mathbb{E}_{(X,Y)\sim\mathcal{P}_{XY}}[\ell(f_{\text{SDRM}}(X), Y)], \tag{6}$$

*which implies SGDG is expected to show better generalization performance compared to SDRM. By Proposition 1, SGDG is expected to perform better than SDRM for the domain generalization problem equivalently. The sketch of proof of this theorem is that SGDG performs as well as SDRM on the source domains and offers performance improvement on the target domains. Proof of this theorem is given in Appendix A.2.*

In reality, we may have no access to the distribution $\mathcal{P}_X^k$ for all domains ($k = 1, \cdots, L$) but only have access to the source domains' distributions ($k = 1, \cdots, K$). In such cases, we use $\mathcal{P}_X^k$ for $k = 1, \cdots, K$ to learn the selection model $\boldsymbol{g}$ as in Problem (7).

**Assumption 4.** *Let $\boldsymbol{g}^*$ be the optimal selection model that has the minimum expected $\ell_s$,*

$$\boldsymbol{g}^* = (g_1^*, \cdots, g_L^*) = \underset{\boldsymbol{g}}{\operatorname{argmin}} \sum_{k=1}^{L} \mathbb{E}_{X \sim \mathcal{P}_X^k}[\ell_s(\boldsymbol{g}(X), S^k = 1)]P(S^k = 1).$$

*We assume that the first $K$ coordinates $(g_1^*, \cdots, g_K^*)$ minimize $\sum_{k=1}^{K} \mathbb{E}_{X \sim \mathcal{P}_X^k}[\ell_s(\boldsymbol{g}(X), S^k = 1)]P(S^k = 1).$*

Conceptually, Assumption 4 means that the same selection models $g_k$ can be learned by contrasting domain $k$ to the remaining domains in the training data ($K \setminus \{k\}$) as by contrasting domain $k$ to the remaining domains in the population ($L \setminus \{k\}$). Under Assumption 4, Problem (5) reduces to

$$\min_{f \in \mathcal{F}, \boldsymbol{g} \in \mathcal{G}} \sum_{k=1}^{K} \mathbb{E}_{(X,Y) \sim \mathcal{P}_{XY}^k} \left[\Lambda(f(X), \boldsymbol{g}(X); Y, S^k = 1)\right]P(S^k = 1) + \sum_{k=1}^{K} \mathbb{E}_{X \sim \mathcal{P}_X^k} \left[\ell_s(\boldsymbol{g}(X), S^k = 1)\right]P(S^k = 1).$$
(7)

# 4 HECKMAN-TYPE SELECTION-GUIDED DOMAIN GENERALIZATION

In this chapter, we present parametric models for $\Lambda$ embodying $f_{\text{SGDG}}$ presented in the previous section. The essence of $\Lambda$ is to model the conditional distribution of $P(Y|X, S^k = 1)$. Consider the setting where we are given training data from multiple source domains in the form of $\mathcal{D} = \{\boldsymbol{x}_i, \boldsymbol{s}_i, y_i\}_{i=1}^N$, where $\boldsymbol{s}_i = [s_{i1}, s_{i2}, \ldots, s_{ik}, \ldots, s_{iK}] \in \{0, 1\}^K$ is a binary vector indicating domain membership. By formulating the loss function as the negative log-likelihood, the empirical form of Equation 7 becomes

$$\min -\sum_{i=1}^{N} \sum_{k=1}^{K} \left[s_{ik} \log p(y_i|\boldsymbol{x}_i, s_{ik} = 1) + s_{ik} \log p(s_{ik} = 1|\boldsymbol{x}_i) + (1 - s_{ik}) \log p(s_{ik} = 0|\boldsymbol{x}_i)\right].$$ (8)

Heckman (1979) proposed to model the joint distribution $P(Y, \tilde{S}|X)$ of the selection latent variable $\tilde{S}$, where $S = \mathbb{I}[\tilde{S} > 0]$, and the continuous outcome $Y$ via a bivariate normal distribution with a correlation coefficient $\rho$ and the mean as a linear functions of $X$. Building upon his work, we make the following assumption on the joint distribution of the outcome and the selection variables.

**Assumption 5** (Joint Distribution of $Y$ (or latent variables $\tilde{Y}$) and $\tilde{S}$). *Let $Y$ be the outcome variable and $\tilde{S}^k$ be the latent continuous variable where $S^k = \mathbb{I}[\tilde{S}^k > 0]$ for each domain $k$. We assume that $Y$ and $S = (S^1, \cdots, S^K)$ are jointly distributed as a multivariate normal distribution with mean $(f(X), g_k(X))$, and correlation coefficients $\{\rho_k\}_{k=1}^K$, given $X$.*

For binary and multinomial outcomes, this assumption is on $\tilde{Y}$ as the latent continuous variables underlying the observed outcomes. Under this assumption, $\mathbf{g} = \{g_k\}_{k=1}^K$ can be modeled as a set of independent probit models. In all three cases, the joint log-likelihoods have closed forms. Henceforth, we refer to the specific parametric form of $f_{\text{SGDG}}$ as the Heckman-type DG (HeckmanDG) estimator.

**Definition 4** (Heckman-Type Domain Generalization Estimator). *We formulate HeckmanDG as a joint learning problem of $f$ and $\boldsymbol{g} = \{g_k\}_{k=1}^K$ with the following learning objective:*

$$\min_{f, \boldsymbol{g}, \Sigma} \sum_{i=1}^{N} \sum_{k=1}^{K} \left[s_{ik} \Lambda\left(f(\boldsymbol{x}_i), g_k(\boldsymbol{x}_i); y_i, s_{ik}; \Sigma\right) - \left\{s_{ik} \log \Phi(g_k(\boldsymbol{x}_i)) + (1 - s_{ik}) \log \Phi(-g_k(\boldsymbol{x}_i))\right\}\right]$$ (9)

*where $\Phi(\cdot)$ is the cumulative distribution function of the standard normal distribution $\phi$, such that $\Phi(g_k(\boldsymbol{x}_i)) = P(S^k = 1|X = \boldsymbol{x}_i)$ is the selection probability w.r.t domain $k$, and $\Phi(-g_k(\boldsymbol{x}_i)) = P(S^k = 0|X = \boldsymbol{x}_i)$. Meanwhile, $\Lambda(f(\boldsymbol{x}_i), g_k(\boldsymbol{x}_i); y_i, s_{ik}; \Sigma)$ is the conditional negative log probability of $y_i$ given $s_{ik} = 1$, i.e., $-\log p(y_i|s_{ik} = 1, \boldsymbol{x}_i)$.*

The specific form of $\Lambda$ and miscellaneous model parameters $\Sigma$ depends on the prediction task (either continuous-valued, binary, or multinomial outcome prediction). For example, the objective function (9) for the continuous outcome is listed below in Equation 10, where we assume $Y = f(X) + \varepsilon$,

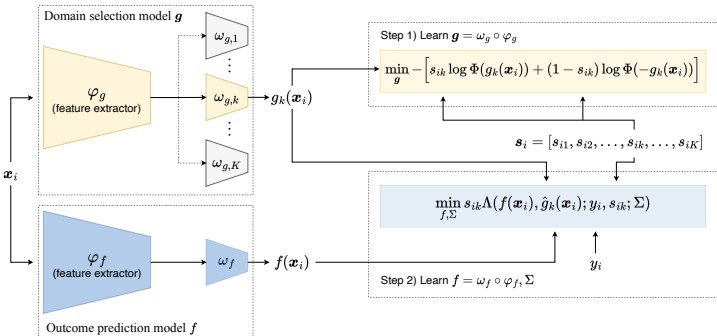

Figure 2: Schematic overview of HeckmanDG, with $K$ training domains. **Step 1)** We learn $\boldsymbol{g} = \{\omega_{g,k} \circ \varphi_g\}_{k=1}^K$ to predict the selection probabilities for each training domain. **Step 2)** The estimated domain selection model $\hat{\boldsymbol{g}}$ guides $f = \omega_f \circ \varphi_f$ to be corrected by considering the selection probability of instances being drawn from the training domains.

$S^k = \mathbb{I}[g_k(X) + \eta_k > 0]$, and $[\eta_k, \varepsilon] \sim \mathcal{N}(0, [1, \sigma; \rho_k])$. The details of its derivation and for other types of outcomes can be found in Appendix A.3.

$$\min_{f,\boldsymbol{g},\Sigma} - \sum_{i=1}^{N} \sum_{k=1}^{K} s_{ik} \left( \log \Phi \left( \rho_k \frac{y_i - f(\boldsymbol{x}_i)}{\sigma} + g_k(\boldsymbol{x}_i) \right) + \log \frac{\phi\left(\frac{y_i - f(\boldsymbol{x}_i)}{\sigma}\right)}{\sigma} \right) + (1 - s_{ik}) \log \Phi(-g_k(\boldsymbol{x}_i)).$$
(10)

Ultimately, $\hat{f}$ is the outcome prediction model of primary interest. For any input $\boldsymbol{x}$ from unseen domains, the HeckmanDG prediction is $\hat{f}(\boldsymbol{x})$.

The proposed HeckmanDG differentiates from Heckman's bias correction method in the following perspectives. *First*, it allows and models multiple domain-specific sample selection mechanisms. *Second*, by utilizing the multiple domains in the training data, HeckmanDG does not need auxiliary data (features of instances from the target population but not sampled in the domain), which is a required input for the original Heckman model. *Third*, HeckmanDG alleviates the linear assumptions in Heckman's bias correction model to allow flexible forms for both prediction and selection functions including neural networks.

## 5 OPTIMIZATION

In this section, we consider the hypothesis sets $\mathcal{F}$ and $\mathcal{G}$ are neural networks, and present an efficient algorithm to optimize Equation 9. Specifically, we propose a two-step approach that primarily trains $\boldsymbol{g}$ to optimum (**Step 1**), then updates the remaining parameters of $f$ and $\Sigma$ (**Step 2**). In Step 1, we learn the selection model parameters based on the following objective function:

$$\hat{\boldsymbol{g}} = \operatorname*{argmin}_{\boldsymbol{g}} - \sum_{i=1}^{N} \sum_{k=1}^{K} \left[ s_{ik} \log \Phi(g_k(\boldsymbol{x}_i)) + (1 - s_{ik}) \log \Phi(-g_k(\boldsymbol{x}_i)) \right]$$
(11)

which essentially learns to predict the source domain memberships for training instances. In Step 2, we freeze the selection model $\hat{\boldsymbol{g}}$, and learn the remaining parameters of $f$ and $\Sigma$. The proposed optimization algorithm is in part motivated by Heckman's two-step estimator for bias correction, which was devised as a means of avoiding the non-linearity of estimating both selection and outcome equations simultaneously (Heckman, 1979). We provide pseudocode describing the overall optimization procedure in Algorithm 1 in Appendix A.4, and an ablation study in Section 7 to support the necessity of two-step optimization.

**Neural Network Architecture.** We use a common feature extractor $\varphi_g : \mathcal{X} \to \mathcal{Z}_g$ for the domain selection models $\boldsymbol{g} = \{g_k\}_{k=1}^K$, which only differ in the last linear predictors $\omega_{g,k} : \mathcal{Z}_g \to [0,1]$, thus $g_k = \omega_{g,k} \circ \varphi_g$. This prevents the number of parameters of **g** from growing with the number of training domains $K$, and reduces computational complexity by requiring only a single forward pass $\varphi_g(\cdot)$. Similarly, we define $f = \omega_f \circ \varphi_f$, where $\varphi_f : \mathcal{X} \to \mathcal{Z}_f$ is the feature extractor of the outcome prediction model and $\omega_f : \mathcal{Z}_f \to \mathcal{Y}$ is the final linear predictor. In general, $\varphi_g$ and $\varphi_f$ are allowed to have different neural architectures with different numbers of parameters. However, we found that simply using the same architecture (but learning different parameters) works well in practice. Therefore, in all our experiments, HeckmanDG has roughly twice as much trainable parameters as other comparative methods. An overview of our neural network architecture is provided in Figure 2.

|  | Train | ID | OOD | Random |
|---|---|---|---|---|
| ERM (oracle) | 1.57 (0.46) | 1.30 (0.49) | 1.30 (0.40) | 1.01 (0.04) |
| ERM | 0.84 (0.09) | 0.89 (0.12) | 14.65 (6.00) | 6.84 (1.83) |
| IRM | 0.83 (0.09) | 0.88 (0.11) | 16.45 (6.40) | 7.42 (1.86) |
| GroupDRO | 0.91 (0.39) | 0.88 (0.12) | 15.83 (5.22) | 7.33 (1.67) |
| VREx | 0.85 (0.08) | 0.89 (0.13) | 14.47 (4.91) | 6.96 (1.68) |
| **HeckmanDG** (ours) | 1.40 (0.24) | 1.37 (0.29) | **8.71** (3.96) | **3.70** (0.96) |

Table 1: Predictive performance on simulated data, measured in terms of the mean squared error (*lower is better*) averaged over 30 trials. Standard deviations are given in parentheses.

## 6 EXPERIMENTS

**Simulation.** We simulate a linear regression problem to assess HeckmanDG's predictive performance. We simulate two covariates $(X_1, X_2)$ and two training domains $(K = 2)$ based on the following setting of domain selection and outcome mechanisms:

$$S^k = \mathbb{I}[\alpha_0^k + \alpha_2^k X_2 + \eta_k > 0], \quad Y = 1 + 1.5X_1 + 3X_2 + \varepsilon$$
$$\begin{bmatrix} X_1 \\ X_2 \end{bmatrix} \sim \mathcal{N}(\mathbf{0}, \mathbf{I}_2), \text{ and } \begin{bmatrix} \eta_k \\ \varepsilon \end{bmatrix} \sim \mathcal{N}\left(\begin{bmatrix} 0 \\ 0 \end{bmatrix}, \begin{bmatrix} 1 & \rho_k\sigma \\ \rho_k\sigma & \sigma^2 \end{bmatrix}\right) \quad (12)$$

where we assume a normal prior over the selection coefficient $\alpha_2^k \sim \mathcal{N}(\mu_{\alpha_2}, \sigma_{\alpha_2}^2)$ in order to implicitly control the similarity between domains by differing its parameters. We consider a true population of 100000 instances, from which we sample data for each domain with $n_k \sim \text{Uniform}(1000, 2000)$. In each trial, we sample the selection coefficient $\alpha_2^k \sim \mathcal{N}(5, 3^2)$ for the two training domains and a held-out in-distribution (ID) test set (similar to the training domains), and $\alpha_2^{k'} \sim \mathcal{N}(-5, 3^2)$ for another held-out out-of-distribution (OOD) test set (dissimilar to the training domains). We also simulate a random test set from the population. We assumed $\rho_k = 0.8$ and $\sigma = 1$.

**Simulation Results.** We observed that HeckmanDG not only outperforms ERM, but also other DG methods including IRM (Arjovsky et al., 2019), GroupDRO (Sagawa et al., 2020), and VREx (Krueger et al., 2021) (Table 1). Note that 'ERM (oracle)' is trained on *iid* training data, which serves as a theoretical lower bound on model performance. We highlight that not only does HeckmanDG perform well on the random test set, but also on the OOD test set. In contrast, other methods tend to fit well to the train domains (ID performance is high), but generalize poorly to random and OOD test domains.

**Benchmark Datasets.** To further demonstrate the effectiveness of HeckmanDG on high-dimensional data regimes, we conducted experiments on four datasets from the WILDS benchmark (Koh et al., 2021): 1) CAMELYON17, 2) POVERTYMAP, 3) IWILDCAM, and 4) RXRX1. We used the same neural network architecture for selection and outcome feature extractors $\varphi_g$ and $\varphi_f$, which are followed by linear predictors $\omega_g$ and $\omega_f$. For the domain selection model, we tuned hyperparameters to obtain the best domain selection model returning the highest macro F1 score on the training data, which in all cases achieve near-perfect accuracy. For the outcome prediction model, we adhered to the official guidelines and used the OOD validation set provided in the WILDS repository for hyperparameter tuning and model selection based on the recommended metrics. Detailed descriptions of dataset statistics are presented in Table 5 of Appendix A.4. Details on hyperparameters and model training are presented in Table 6 of Appendix A.4.

**Benchmark Datasets Results.** We summarized the results on the WILDS benchmark in Tables 2, 3, and 4. All methods apply the same network architectures based on the WILDS guideline for fair comparison. Also, we point out that we exclude methods that deviate from the DG setting such as those centered on test time adaptation and using additional unlabeled data. We observed that HeckmanDG outperforms other methods on two out of four datasets, while performing on par with other methods on the remaining two. We make the following key observations. *First*, HeckmanDG often robustly performs on the test domain although it may not generate the best performance on the validation domains. For example, on the CAMELYON17 dataset, the performance gap between the validation and test datasets for HeckmanDG is 3.3, which is substantially smaller than 14.1 of

| Method | Validation | Test |
|---|---|---|
| ERM (scratch) | 84.9 (3.1) | 70.8 (7.2) |
| ERM (ImageNet) | **91.3** (0.2) | 84.2 (1.7) |
| CORAL | 86.2 (1.4) | 59.5 (7.7) |
| IRM | 86.2 (1.4) | 64.2 (8.1) |
| GroupDRO | 85.5 (2.4) | 68.4 (7.3) |
| VREx | 82.3 (1.3) | 71.5 (8.3) |
| LISA | 81.8 (1.3) | 77.1 (6.5) |
| Fish | 82.5 (1.2) | 79.5 (6.0) |
| SWAD | 88.1 (1.5) | 83.9 (0.9) |
| L2A-OT | 86.3 (3.4) | 77.5 (6.7) |
| **HeckmanDG** (ours) | 90.6 (2.4) | **87.3** (2.4) |

Table 2: CAMELYON17: We report predictive performance measured in terms of average accuracy on both the OOD validation and test set. Standard deviation across 10 replicates are given in parentheses. 'ERM (scratch)' is trained from random initial parameters, whereas we also report 'ERM (ImageNet)' trained from ImageNet-pretrained weights (Russakovsky et al., 2015).

| | Average | | Worst Group | |
|---|---|---|---|---|
| Method | Validation | Test | Validation | Test |
| ERM | 0.80 (0.04) | 0.78 (0.03) | 0.51 (0.06) | 0.45 (0.06) |
| CORAL | 0.80 (0.04) | 0.77 (0.05) | 0.52 (0.06) | 0.44 (0.06) |
| IRM | **0.81** (0.03) | 0.77 (0.05) | **0.53** (0.05) | 0.43 (0.07) |
| GroupDRO | 0.78 (0.05) | 0.75 (0.07) | 0.46 (0.04) | 0.39 (0.06) |
| DANN | 0.77 (0.04) | 0.69 (0.04) | 0.44 (0.11) | 0.33 (0.10) |
| Fish | **0.81** (0.01) | **0.81** (0.01) | - | - |
| SWAD | 0.78 (0.03) | 0.77 (0.04) | 0.48 (0.09) | 0.45 (0.11) |
| **HeckmanDG** (ours) | **0.81** (0.03) | **0.81** (0.03) | **0.53** (0.06) | **0.51** (0.04) |

Table 3: POVERTYMAP: We report predictive performance measured in terms of both the *average* and *worst-group* Pearson correlation coefficient, on both the OOD validation and test sets. Standard deviation across 5 replicates are given in parentheses. We use the original 5 folds provided in the WILDS repository. We do not report worst-group performance for 'Fish', since it has not been reported in Shi et al. (2022).

'ERM (scratch)' and 7.1 of 'ERM (ImageNet)'. This pattern is similar to what we observed in the simulation studies. Because HeckmanDG is designed to predict for the underlying population, it may not produce the best prediction performances for the (non-randomly selected) source domains. Although we do not know the specific selection probability of the test domain, HeckmanDG is often more robust for the testing domains by optimizing for the population distribution. *Second*, HeckmanDG effectively works for both the domain shift and the subpopulation shift problems. This can be observed by looking at either the average or worst-group performance on the POVERTYMAP dataset (i.e., 0.51). This again supports the robustness of HeckmanDG against a range of non-random selection probabilities of the testing domains.

## 7 ANALYSIS

**Necessity of two-step optimization.** To demonstrate the necessity and effectiveness of the proposed two-step optimization, we performed an ablation study on the POVERTYMAP dataset. For comparison, we trained $g$ and $f$ in *one-step* to simultaneously minimize Equation 8. We observed that *one-step* obtained comparable performances on training domains (Figure 3a), but yielded worse test performance on the OOD domains (Figure 3b). We believe that a suboptimal $\hat{g}$ may mislead $\hat{f}$ to a deficient solution in the one-step optimization (more details in Appendix A.6).

**Relationship between the performance of $g$ and $f$.** To further investigate how the performance of a domain selection model influences the final outcome prediction, we take intermediate snapshots for $\hat{g}$, and learned $f$ based on each snapshot. On all five data split folds, we always observed a positive

| (a) RxRx1 | | |
| --- | --- | --- |
| Method | Validation | Test |
| ERM | 19.4 (0.2) | 29.9 (0.4) |
| CORAL | 18.5 (0.4) | 28.4 (0.3) |
| IRM | 5.6 (0.4) | 8.2 (1.1) |
| GroupDRO | 15.2 (0.1) | 23.0 (0.3) |
| LISA | 20.1 (0.4) | 31.9 (1.0) |
| Fish | 7.5 (0.6) | 10.1 (1.5) |
| SWAD | 14.2 (0.5) | 22.9 (0.7) |
| L2A-OT | 17.5 (0.3) | 27.8 (0.9) |
| **HeckmanDG** (ours) | **20.5** (0.7) | **32.1** (0.8) |

| (b) iWildCam | | |
| --- | --- | --- |
| Method | Validation | Test |
| ERM | **37.4** (1.3) | 31.0 (1.3) |
| CORAL | 37.0 (1.2) | **32.8** (0.1) |
| IRM | 20.2 (7.6) | 15.1 (4.9) |
| GroupDRO | 26.3 (0.2) | 23.9 (2.1) |
| DANN | - | 31.9 (1.4) |
| Fish | 25.8 (0.5) | 24.2 (0.9) |
| SWAD | 31.6 (0.2) | 29.1 (0.1) |
| L2A-OT | 22.8 (2.9) | 18.1 (3.2) |
| **HeckmanDG** (ours) | 34.5 (0.9) | 31.8 (0.3) |

Table 4: A summary of results on RxRx1 and iWildCam. Note that the OOD validation macro F1 score for DANN was not reported since it has not been provided in Sagawa et al. (2022).

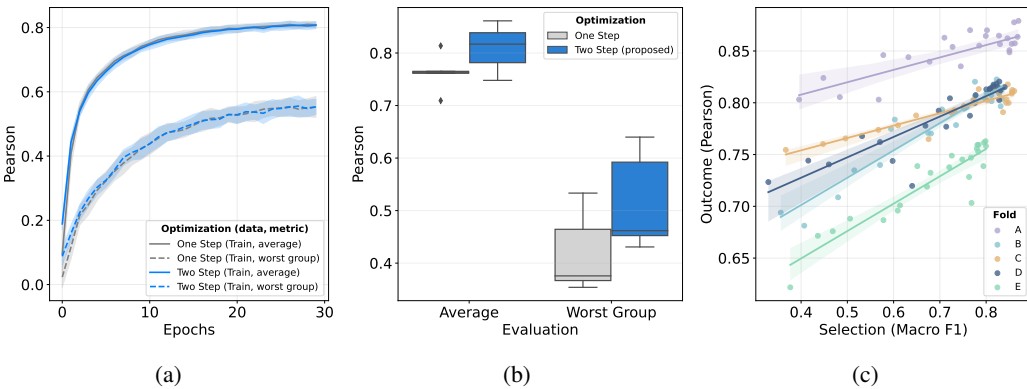

Figure 3: Analysis based on the PovertyMap dataset. 3a) Pearson correlation measured on the training data for HeckmanDG learned by *one-step* (gray) and *two-step* (blue, proposed) optimization. 3b) Comparison between *one-step* (gray) and *two-step* (blue, proposed) optimization, where performance is averaged across all five folds provided in the Wilds repository. 3c) Relationship between the performance of domain selection and outcome prediction (colors indicate different folds).

correlation between the performances of $\hat{g}$ and $\hat{f}$ (Figure 3c), demonstrating that a well-performing selection model is necessary to correctly guide the outcome prediction model.

# 8 Conclusions

We propose a Selection Guided Domain Generalization (SGDG) framework, in which we formulate domain generalization as a non-random sample selection problem and propose to jointly learn the prediction model $f$ and the domain selection model $g$ to achieve generalization on the true population. DG is a challenging problem as the particular structure of the distribution shift in the testing domains is unknown. In the presence of this uncertainty, we propose and theoretically justify the objective of minimizing the risk targeting the population distribution through SGDG. Furthermore, we have provided a set of Heckman-type SGDG estimators for various outcome types under parametric assumptions. Although it is still an open question if a single general-purpose training algorithm can produce models that do well on all of the DG datasets (Koh et al., 2021), we observed robust performances of the proposed HeckmanDG on four benchmark datasets.

Note that SGDG can naturally utilize all domains in the training data, including the (outcome) labeled and unlabeled domains. The unlabeled domains will contribute to the estimation of $\hat{g}$ in Equation 9, which indirectly improves generalization performance as we showed in Section 7. An intriguing direction for future research is to explore whether we can improve SGDG performances by adapting $\hat{g}(X)$ for domains in the test data. In that way, we may further refine the prediction for the unseen domains guided by their similarities to the source domains.

## ACKNOWLEDGEMENT

We thank the ICLR reviewers for their suggestions on improving the original version of this paper. This work was supported in part by the National Institutes of Health under awards NIH R01-LM013344, R01AG054467, R01AG065330, and R01-AG065330-02S1, by NYU Center for the Study of Asian American Health under the NIH/NIMHD grant award #U54MD000538 (HD).

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

# A APPENDIX

## A.1 DERIVATION OF EXAMPLE OF DECOMPOSABLE LOSS

$$
\begin{aligned}
\ell(f(X), Y) &= -\log p(Y|X) \\
&= \sum_{k=1}^{L} -\log p(Y, S^k = 1|X) \\
&= \sum_{k=1}^{L} \Big\{ -\log \big[ p(Y|X, S^k = 1) p(S^k = 1|X) \big] \Big\} \\
&= \sum_{k=1}^{L} \Big\{ -\log p(Y|X, S^k = 1) - \log p(S^k = 1|X) \Big\} \\
&= \sum_{k=1}^{L} \underbrace{-\log p(Y|X, S^k = 1)}_{\Lambda} + \sum_{k=1}^{L} \underbrace{-\log p(S^k = 1|X)}_{\ell_s}.
\end{aligned}
$$

## A.2 PROOF OF THEOREM 1

The following chain of inequalities completes the proof:

$$
\mathbb{E}_{(X,Y) \sim \mathcal{P}_{XY}} [\ell(f_{\text{SDRM}}(X), Y)] = \sum_{k=1}^{L} \mathbb{E}_{(X,Y) \sim \mathcal{P}_{XY}^k} [\ell(f_{\text{SDRM}}(X), Y)] P(S^k = 1)
$$

$$
= \sum_{k=1}^{L} \mathbb{E}_{(X,Y) \sim \mathcal{P}_{XY}^k} [\Lambda(f_{\text{SDRM}}(X), g_{\text{SGDG}}(X); Y, S^k)] + \sum_{k=1}^{L} \mathbb{E}_{X \sim \mathcal{P}_X^k} [\ell_s(g_{\text{SGDG}}(X), S^k = 1)] P(S^k = 1)
$$

$$
\geq \sum_{k=1}^{K} \mathbb{E}_{(X,Y) \sim \mathcal{P}_{XY}^k} [\Lambda(f_{\text{SGDG}}(X), g_{\text{SGDG}}(X); Y, S^k)] + \sum_{k=1}^{L} \mathbb{E}_{X \sim \mathcal{P}_X^k} [\ell_s(g_{\text{SGDG}}(X), S^k = 1)] P(S^k = 1)
$$

$$
+ \sum_{k=K+1}^{L} \mathbb{E}_{(X,Y) \sim \mathcal{P}_{XY}^k} [\Lambda(f_{\text{SDRM}}(X), g_{\text{SGDG}}(X); Y, S^k)] \tag{13}
$$

$$
\geq \sum_{k=1}^{K} \mathbb{E}_{(X,Y) \sim \mathcal{P}_{XY}^k} [\Lambda(f_{\text{SGDG}}(X), g_{\text{SGDG}}(X); Y, S^k)] + \sum_{k=1}^{L} \mathbb{E}_{X \sim \mathcal{P}_X^k} [\ell_s(g_{\text{SGDG}}(X), S^k = 1)] P(S^k = 1)
$$

$$
+ \sum_{k=K+1}^{L} \mathbb{E}_{(X,Y) \sim \mathcal{P}_{XY}^k} [\Lambda(f_{\text{SGDG}}(X), g_{\text{SGDG}}(X); Y, S^k)] \tag{14}
$$

$$
= \mathbb{E}_{(X,Y) \sim \mathcal{P}_{XY}} [\ell(f_{\text{SGDG}}(X), Y)].
$$

Note that 13 follows from the definition of SGDG. 14 holds if $Y$ and $S$ are dependent and if we let $\ell(f(X), Y) = -\log p(Y|X)$, $\Lambda = -\log p(Y|X, S^k = 1)$, and $\ell_s = -\log p(S^k = 1|X)$.

## A.3 DETAILS OF HECKMAN CORRECTION-GUIDED DOMAIN GENERALIZATION

In this section, we present the exact form of our loss functions for continuous, binary, and multinomial outcomes, based on their specific parametric assumptions.

**Continuous Outcomes** For regression tasks where $y \in \mathbb{R}$, we assume that the selection and outcome is generated by the following data generation process:

$$
S^k = \mathbb{I}[\tilde{S}^k > 0] = \mathbb{I}[g_k(\boldsymbol{x}) + \eta_k > 0] \tag{15}
$$

$$
Y = f(\boldsymbol{x}) + \varepsilon \tag{16}
$$

$$\begin{bmatrix} \eta_k \\ \varepsilon \end{bmatrix} \sim \mathcal{N}\left( \mathbf{0}, \begin{bmatrix} 1 & \sigma\rho_k \\ \sigma\rho_k & \sigma^2 \end{bmatrix} \right) \tag{17}$$

where $\mathbb{I}(\cdot) \in \{0,1\}$ is the indicator function, $\rho_k$ is the correlation between $\eta_k$ and $\varepsilon$, and $\sigma \in \mathbb{R}_+$ is the standard deviation of $\varepsilon$. Being consistent with the notation in the main text, we hereby define $\Sigma = \{\rho_1, \ldots, \rho_K, \sigma\}$. Denoting model parameters as $\theta = \{f, g_1, \ldots, g_K, \Sigma\}$, we formulate the data likelihood as follows:

$$\mathcal{L}_c(\theta; \mathcal{D}) = \prod_{i=1}^{N} \prod_{k=1}^{K} \left[ p(y_i|\boldsymbol{x}_i, s_{ik}=1) \cdot p(s_{ik}=1|\boldsymbol{x}_i) \right]^{s_{ik}} \cdot p(s_{ik}=0|\boldsymbol{x}_i)^{1-s_{ik}} \tag{18}$$

and we take the negative log likelihood to formulate the loss function:

$$\begin{aligned}
\ell_c(\mathcal{D}; \theta) = & -\sum_{i=1}^{N}\sum_{k=1}^{K} s_{ik} \left[ \log \Phi_1\left( \frac{g_k(\boldsymbol{x}_i) + \rho_k \frac{y_i - f(\boldsymbol{x}_i)}{\sigma}}{\sqrt{1-\rho_k^2}} \right) - \frac{1}{2}\log 2\pi\sigma^2 - \frac{1}{2}\left\{ \frac{y_i - f(\boldsymbol{x}_i)}{\sigma} \right\}^2 \right] \\
& -\sum_{i=1}^{N}\sum_{k=1}^{K} (1-s_{ik})\log \Phi_1(-g_k(\boldsymbol{x}_i))
\end{aligned}$$
$$\tag{19}$$

**Binary Outcomes**   For binary outcomes where $y \in \{0,1\}$, we assume a probit model for both selection and outcome:

$$\begin{aligned}
S^k &= \mathbb{I}[\tilde{S}^k > 0] = \mathbb{I}[g_k(\boldsymbol{x}) + \eta_k > 0] \\
Y &= \mathbb{I}[f(\boldsymbol{x}) + \varepsilon > 0] \\
\begin{bmatrix} \eta_k \\ \varepsilon \end{bmatrix} &\sim \mathcal{N}\left( \mathbf{0}, \begin{bmatrix} 1 & \rho_k \\ \rho_k & 1 \end{bmatrix} \right)
\end{aligned} \tag{20}$$

where $\rho_k$ is the correlation between $\eta_k$ and $\varepsilon$. Herein, $\Sigma = \{\rho_1, \ldots, \rho_K\}$. Denoting model parameters as $\theta = \{f, g_1, \ldots, g_K, \Sigma\}$, we formulate the data likelihood for binary outcomes as follows:

$$\mathcal{L}_b(\theta; \mathcal{D}) = \prod_{i=1}^{N} \prod_{k=1}^{K} \left\{ p(y_i=1, s_{ik}=1|\boldsymbol{x}_i)^{y_i} \cdot p(y_i=0, s_{ik}=1|\boldsymbol{x}_i)^{1-y_i} \right\}^{s_{ik}} p(s_{ik}=0|\boldsymbol{x}_i)^{1-s_{ik}} \tag{21}$$

and take the negative log likelihood to define the loss function for binary outcomes:

$$\begin{aligned}
\ell_b(\mathcal{D}; \theta) = & -\sum_{i=1}^{N}\sum_{k=1}^{K} s_{ik} \cdot y_i \cdot \log \Phi_2\left( g_k(\boldsymbol{x}_i), f(\boldsymbol{x}_i), \rho_k \right) \\
& -\sum_{i=1}^{N}\sum_{k=1}^{K} s_{ik} \cdot (1-y_i) \cdot \log\left( \Phi_1\left( g_k(\boldsymbol{x}_i) \right) - \Phi_2\left( g_k(\boldsymbol{x}_i), f(\boldsymbol{x}_i); \rho_k \right) \right) \\
& -\sum_{i=1}^{N}\sum_{k=1}^{K} (1-s_{ik}) \cdot \log \Phi_1\left( -g_k(\boldsymbol{x}_i) \right)
\end{aligned} \tag{22}$$

where $\Phi_2(\cdot, \cdot; a)$ is the cumulative density function of the bivariate standard normal given correlation $a \in [-1,1]$.

**Multinomial Outcomes**   For multinomial outcome tasks where $y \in \{1, \ldots, J\}$, we assume a multinomial probit outcome model (McFadden, 1989):

$$\begin{aligned}
S^k &= \mathbb{I}[\tilde{S}^k > 0] = \mathbb{I}[g_k(\boldsymbol{x}) + \eta_k > 0] \\
\tilde{Y}_j &= f_j(\boldsymbol{x}) + \varepsilon_j, \quad \forall j \in \{1, \ldots, J\} \\
Y &= \operatorname*{argmax}_{j \in \{1, \ldots, J\}} \tilde{Y}_j \\
\begin{bmatrix} \eta_k \\ \varepsilon_j \end{bmatrix} &\sim \mathcal{N}\left( \mathbf{0}, \begin{bmatrix} 1 & \rho_k \\ \rho_k & 1 \end{bmatrix} \right)
\end{aligned} \tag{23}$$

where $[\eta_k, \varepsilon_j] \sim \mathcal{N}(\mathbf{0}, [1, 1; \rho_{kj}])$ and $\rho_{kj}$ is the correlation between $\rho_k$ and $\varepsilon_j$. We further assume that the outcome error terms are independently distributed: $\varepsilon = [\varepsilon_1, \ldots, \varepsilon_J]^\top \sim \mathcal{N}(\mathbf{0}, \mathbf{I})$. Consequently, $\Sigma = \{\rho_{11}, \ldots, \rho_{KJ}\}$. Denoting model parameters as $\theta = \{f_1, \ldots, f_J, g_1, \ldots, g_K, \rho_{11}, \ldots, \rho_{KJ}\}$, we formulate the multinomial outcome data likelihood as follows:

$$\mathcal{L}_m(\theta; \mathcal{D}) = \prod_{i=1}^{N} \prod_{k=1}^{K} \prod_{j=1}^{J} \left\{ p(s_{ik} = 1, y_i = j | \boldsymbol{x}_i)^{\mathbb{I}[y_i = j]} \right\}^{s_{ik}} p(s_{ik} = 0 | \boldsymbol{x}_i)^{1 - s_{ik}} \quad (24)$$

where $\mathbb{I}[\cdot] \in \{0, 1\}$ is the indicator function. We use the negative logarithm of the data likelihood for the loss function:

$$\ell_m(\mathcal{D}; \theta) = -\sum_{i=1}^{N} \sum_{k=1}^{K} \sum_{j=1}^{J} \left[ s_{ik} \cdot \mathbb{I}[y_i = j] \cdot \log \int_T \phi(\boldsymbol{u} | \vec{0}, \tilde{\Sigma}) d\boldsymbol{u} + (1 - s_{ik}) \cdot \log \Phi(-g_k(\boldsymbol{x}_i)) \right] \quad (25)$$

where $T = [-g_k(\boldsymbol{x}_i), \infty) \times [\xi_{j,1}(\boldsymbol{x}_i), \infty) \times \ldots \times [\xi_{j,j-1}(\boldsymbol{x}_i), \infty) \times [\xi_{j,j+1}(\boldsymbol{x}_i), \infty) \times \ldots \times [\xi_{j,J}(\boldsymbol{x}_i), \infty) \subseteq \mathbb{R}^J$ is the half-open $J$-dimensional hyperrectangular domain and $\xi_{j,j'}(\boldsymbol{x}_i) = -f_j(\boldsymbol{x}_i) + f_{j'}(\boldsymbol{x}_i)$. We use the GHK algorithm (Hajivassiliou & Ruud, 1994) to compute the multivariate normal integrals.

## A.4 Optimization, Datasets, and Model Training

---

**Algorithm 1** Two-Step Optimization for HeckmanDG

---

1: **Input:** Data $\mathcal{D} = \{(\boldsymbol{x}_i, y_i, \boldsymbol{s}_i) \in \mathcal{X} \times \mathcal{Y} \times \mathcal{S} : i = 1, \cdots, N\}$, Batch size $B$, Learning rate $\gamma$.
2: **Output:** $\hat{f}, \hat{\boldsymbol{g}} = \{\hat{g}_k\}_{k=1}^K$, and $\hat{\Sigma}$.
3: **Initialize:** $f, \boldsymbol{g}, \Sigma$
4: **Step 1:** Learn the Domain Selection Models ($\boldsymbol{g}$)
5:     **for all** $k = 1, \ldots, K$ **do**
6:         $\mathcal{D}_k \leftarrow \{(\boldsymbol{x}_i, s_{ik}) \in \mathcal{X} \times [0, 1] : i = 1, \ldots, N\}$
7:     **end for**
8:     **while** $\boldsymbol{g} = \{g_k\}_{k=1}^K$ `Not Converged` **do**
9:         **for all** $k = 1, \cdots, K$ **do**
10:             $\mathcal{B}_k \leftarrow$ `BatchSampler`$(\mathcal{D}_k, B)$
11:             $g_k \leftarrow g_k + \frac{\gamma}{B} \nabla \sum_{(\boldsymbol{x}_i, s_{ik}) \in \mathcal{B}_k} [s_{ik} \log \Phi(g_k(\boldsymbol{x}_i)) + (1 - s_{ik}) \log \Phi(-g_k(\boldsymbol{x}_i))]$
12:         **end for**
13:     **end while**
14:     $\hat{g}_k = g_k$, for $k = 1, \ldots, K$.
15: **Step 2:** Learn the Outcome Model ($f, \Sigma$)
16:     $\mathcal{D}_{\mathcal{O}} = \mathcal{D}$
17:     **while** $f$ `Not Converged` **do**
18:         $\mathcal{B} =$ `BatchSampler`$(\mathcal{D}_{\mathcal{O}}, B)$
19:         $f \leftarrow f - \frac{\gamma}{B} \nabla \sum_{(\boldsymbol{x}_i, y_i, \boldsymbol{s}_i) \in \mathcal{B}} \sum_{k=1}^{K} s_{ik} \Lambda(f(\boldsymbol{x}_i), \hat{g}_k(\boldsymbol{x}_i); y_i, s_{ik}; \Sigma)$
20:         $\Sigma \leftarrow \Sigma - \frac{\gamma}{B} \nabla \sum_{(\boldsymbol{x}_i, y_i, \boldsymbol{s}_i) \in \mathcal{B}} \sum_{k=1}^{K} s_{ik} \Lambda(f(\boldsymbol{x}_i), \hat{g}_k(\boldsymbol{x}_i); y_i, s_{ik}; \Sigma)$
21:     **end while**
22:     $\hat{f} \leftarrow f$
23:     $\hat{\Sigma} \leftarrow \Sigma$
24: **return** $\hat{f}$

---

| Dataset | CAMELYON17 | POVERTYMAP | IWILDCAM | RXRX1 |
|---|---|---|---|---|
| $X$ | $3 \times 96 \times 96$ (tissue slide) | $8 \times 224 \times 224$ (satellite image) | $3 \times 448 \times 448$ (photo) | $3 \times 256 \times 256$ (cell) |
| $Y$ | 2 (tumor) | continuous (asset wealth) | 182 (animal species) | 1139 (genetic treatments) |
| Training examples | 302,436 | 10,000 | 129,809 | 40,612 |
| Domain | 5 Hospitals (3, 1, 1) | 23 countries (13, 5, 5) | 323 camera traps (243, 32, 48) | 51 batches (33, 4, 14) |
| Evaluation metric | Average accuracy | Pearson (average, worst-group) | Macro F1 | Average accuracy |

Table 5: A summary on the four datasets from WILDS benchmark. In the *'Domain'* row, the three numbers in parentheses denote the number of train, validation, and test domains.

| Dataset | CAMELYON17 | POVERTYMAP | IWILDCAM | RXRX1 |
|---|---|---|---|---|
| Feature Extractor | DenseNet-121 | ResNet-18-MS | ResNet-50 | ResNet-50 |
| Epochs | 5, 5 | 100, 100 | 12, 12 | 10, 10 |
| Batch Size | 32, 32 | 64, 64 | 16, 16 | 75, 75 |
| Optimizer | Adam, SGD | Adam, SGD | Adam, SGD | Adam, SGD |
| Learning Rate | $10^{-5}, 10^{-3}$ | $10^{-5}, 10^{-3}$ | $10^{-5}, 10^{-3}$ | $10^{-5}, 10^{-3}$ |
| Weight Decay | $0, 10^{-4}$ | $0, 10^{-5}$ | $0, 10^{-5}$ | $0, 10^{-5}$ |
| ImageNet Weights | True, True | False, False | True, True | True, True |
| Data Augmentation | N/A | Color jittering | RandAugment | RandAugment |

Table 6: Details on training configurations of HeckmanDG. Cells with two entries (i.e., learning rate, weight decay) denote that we used different values for training domain selection and outcome models. Hyperparameters were determined through grid search on the OOD validation set. For optimizers we searched between $\{\text{Adam}, \text{SGD}\}$. Learning rates were searched among $\{10^{-5}, 10^{-4}, 10^{-3}\}$, and weight decay among $\{0, 10^{-5}, 10^{-3}, 10^{-1}\}$. Note that since HeckmanDG training is two-phase, we searched for hyperparameters sequentially. On all datasets, $\Sigma$ is optimized with the Adam optimizer with a learning rate of $10^{-2}$ and no weight decay.

## A.5 IMPLEMENTATION DETAILS FOR SWAD AND L2A-OT

In Tables 2, 3, and 4, the numbers for SWAD (Cha et al., 2021) and L2A-OT (Zhou et al., 2020) are reproduced based on our implementation. For SWAD, we set the optimum patient parameter $N_s = 3$, the overfitting patient parameter $N_e = 6$, and the tolerance rate $r = 1.2$, which are the default values used in the original paper. The evaluation frequency was set to 100. On all four datasets, models were trained with SGD, using the same learning rates and weight decay factors as ERM (Koh et al., 2021). For L2A-OT, we set $\lambda_{\text{Domain}} = 0.5$, $\lambda_{\text{Cycle}} = 10$, and $\lambda_{\text{CE}} = 1$. On all three datasets (excluding POVERTYMAP), the generator $G$ is trained with Adam using a constant learning rate of $3 \cdot 10^{-4}$, while the prediction model $F$ is trained with SGD, using the same learning rates and weight decay factors as ERM (Koh et al., 2021). We faithfully refer the readers to the original papers for the details regarding the hyperparameters.

## A.6 ADDITIONAL ANALYSIS ON TWO-STEP OPTIMIZATION

We provide further details on the analysis in Section 7, conducted to demonstrate the necessity and effectiveness of the proposed two-step optimization approach. For both one-step (jointly optimizing for Equation 8) and two-step, we kept the number of training epochs to 30, with the same batch size, in order to make a valid comparison. In Figure 4, we plot the training loss trajectories using different metrics. We observed that $\mathbf{g}$ tends to converge to a suboptimal point if we optimize $f$, $\mathbf{g}$, and $\Sigma$ simultaneously by a one-step optimization approach (Figure 4a). We suspect that this happens because the gradient of $\mathbf{g}$ (involving $s_{ik} = 1$) depends on $\Sigma$, which changes during the joint one-step optimization process. This will often result in a suboptimal $\hat{\mathbf{g}}$, as shown in Figure 4d. Note that the original Heckman correction (Heckman, 1979) also proposed a two-step approach to avoid the computational burden of having to estimate both $f$ and $g$ jointly. One way to avoid this problem is to use an alternating optimization algorithm which alternately minimizes Equation 8 with respect to $\mathbf{g}$, $\Sigma$, and $f$ until convergence. Our two-step approach can be regarded as a one-iteration alternating minimization procedure as it stops after a single iteration. In Figure 4c, the one-step method yields comparable (MSE) loss values for $f$ as those obtained from the two-step method on the training dataset. However Equation 8 converged to a larger value by the one-step method, implying that it converged to a suboptimal point for $\mathbf{g}$. On the test datasets, the suboptimal solution of the one-step method resulted in worse domain generalization performance compared to the two-step method (Figure 3b).

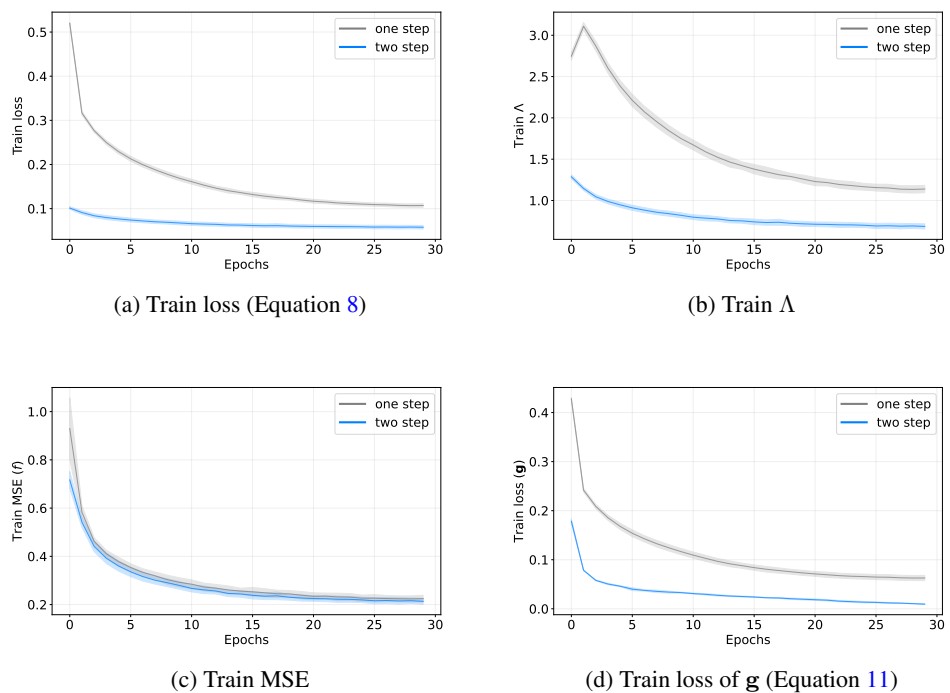

(a) Train loss (Equation 8)

(b) Train $\Lambda$

(c) Train MSE

(d) Train loss of $\mathbf{g}$ (Equation 11)

Figure 4: Training loss of one-step (gray) vs. two-step (blue) optimization on POVERTYMAP.

