# OpenReview forum: "Domain Generalization via Heckman-type Selection Models "
_ICLR.cc/2023/Conference — ICLR 2023 notable top 25%_

### Official Review · Reviewer_CmDk · 2022-10-23

**Confidence:** 3
**Correctness:** 3
**Technical Novelty And Significance:** 4
**Empirical Novelty And Significance:** 3
**Recommendation:** 6

**Clarity, Quality, Novelty And Reproducibility:**

It was not very easy for me to follow the paper. I barely understood the abstract and the introduction before going to the technical details. In contrast, it was a relatively smooth read after Section 3. I list some specific points below that are unclear to me:
- What does it mean by "non-random"? The proposed method treats all the variables as random variables. It sounds contradictory to me.
- "A naively trained model, optimizing the prediction performance conditional on the selection mechanisms of the domains in the training data, will fail to perform similarly if sample selection of a testing domain is under a different selection mechanism.": I could not understand this sentence.
- "In such cases, we use $P^k_X$ for $k = 1, \dots, K$ as a surrogate, implying that we learn selection models by contrasting one domain to the remaining source domains": This is not very clear to me. It would be nice if there were more explanations since, I suppose, this is part of the proposed method.


**Strength And Weaknesses:**

# Strengths
- The approach of exploiting the correlation between the data and the domain is interesting.
- Theorem 1 is also interesting and well supports the proposed approach.
- The experiments show that the proposed method significantly outperforms previous approaches.

# Weaknesses
- In Section 4, "Equation 5 becomes...": I think this is related to the previous point. Eq. (7) does not exactly correspond to Eq. (5) because it ignores $k = K + 1, \dots, L$. But if we ignore those terms, we will go back to Eq. (3). I am not sure if the proposed method is doing what the paper claims it does.
- The two-step algorithm does not exactly solve Eq. (8) but performs better than the more theoretically faithful one-step algorithm, according to Section 7. The paper does not provide a convincing explanation about this. I suspect that the two-step algorithm is solving something different from Eq. (8) but theoretically justifiable.

**Summary Of The Paper:**

This paper proposes a method for domain generalization in which the variable $S$ selecting the domain correlates both the input $X$ and the output variable $Y$. The proposed method models the joint probability distribution of $(X, Y, S)$ and performs a two-step variant of the maximum likelihood estimation (MLE). The authors demonstrate that the proposed method outperforms the ERM as well as other domain generalization methods. Interestingly, the proposed two-step algorithm performs better than jointly performing the MLE.

**Summary Of The Review:**

The proposed method is interesting and seemingly effective, but it looks like there is a gap between the theory part and the proposed algorithm. I weakly recommend acceptance.

---

> ### Author Response · Authors · 2022-11-18
> **Thank you for the insightful comments. Here is the response to Reviewer CmDk (1/2).**
>
> > Q1) In Section 4, "Equation 5 becomes...": I think this is related to the previous point. Eq. (7) does not exactly correspond to Eq. (5) because it ignores k=K+1,…,L. But if we ignore those terms, we will go back to Eq. (3). I am not sure if the proposed method is doing what the paper claims it does.
>
> A1) We apologize for this confusion, and agree that we need to be more clear and rigorous for connecting Eq. (5) and Eq. (7). As pointed out, Eq. (7) does not correspond to Eq. (5) in general. Eq. (7) is a realization of:
>
> \begin{equation} \tag{R.1}
>   \underset{f \in \mathcal{F}, \boldsymbol{g} \in \mathcal{G}}{\min} \Bigg[
>     \sum\_{k=1}^{K} \underset{X,Y \sim \mathcal{P}^{k}_{XY}}{\mathbb{E}}\Big[ \Lambda(f(X),\boldsymbol{g}(X);Y,S^{k}=1) \Big] P(S^{k}=1) +\sum\_{k=1}^{K} \underset{X \sim \mathcal{P}\_{X}^{k}}{\mathbb{E}}\Big[ \ell\_{s}(\boldsymbol{g}(X),S^{k}=1) \Big] P(S^{k}=1)
>   \Bigg]
> \end{equation}
>
> which is Eq. (5) without $k=K+1,\cdots,L$ in the second term. Under Assumption 4 (now presented clearly in the revised manuscript), this is a reasonable approximation of Eq. (5). Conceptually, Assumption 4 means that the same selection models $g^k$ can be learned by contrasting domain $k$ to the remaining domains in the training data ($K \setminus \lbrace k \rbrace$) as by contrasting domain $k$ to the remaining domains in the population ($L\setminus \lbrace k \rbrace$). We previously described this assumption in words (top paragraph on page 5). We stated this assumption and Eq. (R.1) in the revised manuscript. We further note that Eq. (R.1) is different from Eq. (3), which is the source domain risk minimization problem:
>
> \begin{equation}\nonumber
>   \min\_{f \in \mathcal{F}} \sum\_{k=1}^{K}\underset{(X,Y) \sim \mathcal{P}^{k}_{XY}}{\mathbb{E}}\Big[ \ell(f(X), Y)\Big]P(S^{k}=1).
> \end{equation}
>
> The two problems become identical only if $\mathcal{P}\_{XY}^{k} = \mathcal{P}\_{XY}$ for all $k=1,\cdots,K$ (because $\mathbb{E}\_{(X,Y) \sim \mathcal{P}\_{XY}^{k}}[\ell(f(X),Y)] \neq \mathbb{E}\_{(X,Y) \sim \mathcal{P}\_{XY}^{k}}[\Lambda(f(X),\boldsymbol{g}(X);Y,S^{k}=1)] + \mathbb{E}\_{X \sim \mathcal{P}\_{X}^{k}}[\ell\_{s}(\boldsymbol{g}(X),S^{k}=1)]$.)
>
> > Q2) The two-step algorithm does not exactly solve Eq. (8) but performs better than the more theoretically faithful one-step algorithm, according to Section 7. The paper does not provide a convincing explanation about this. I suspect that the two-step algorithm is solving something different from Eq. (8) but theoretically justifiable.
>
> A2) We thank you for this suggestion and have provided a more thorough explanation in Section 7. We observed that $\boldsymbol{g}$ tends to converge to a suboptimal point if we optimize $f$, $\boldsymbol{g}$, and $\Sigma$ in Eq. (8) simultaneously by a one-step optimization approach. We suspect that it is because the gradient of $\boldsymbol{g}$ (the part involving $s\_{ik} = 1$) depends on $\Sigma$, which changes during the joint optimization process. This often results in a suboptimal $\hat{\boldsymbol{g}}$ from the one-step optimization. Note that in the original Heckman paper, a two-step approach was also proposed to avoid the computational burden of having to estimate both $f$ and $g$ jointly [1,2,3]. One way to avoid this problem is an alternating minimization algorithm which alternately minimizes Eq. (8) with respect to $\boldsymbol{g}$, $\Sigma$, and $f$ until convergence. Our two-step approach can be regarded as a one-iteration alternating minimization approach as it stops after one iteration. As presented in the figures in the Appendix A.5, for PovertyMap, the one-step method provided comparable (MSE) loss values for $f$ as those obtained from the two-step methods on the training dataset. However, Eq. (8) converged to a bigger value by the one-step method, implying that it converged to a suboptimal point for $\boldsymbol{g}$. On the testing datasets, the suboptimal solution from the one-step method resulted in worse domain generalization performance than the two-step method (Figure 3-b). A future direction is to compare the two-step optimization results with the alternating minimization approach (computationally demanding).
>
> [1] Heckman, J. J. (1976). The common structure of statistical models of truncation, sample selection and limited dependent variables and a simple estimator for such models. In Annals of economic and social measurement, volume 5, number 4 (pp. 475-492). NBER.
>
> [2] Heckman, J. J. (1979). Sample selection bias as a specification error. Econometrica: Journal of the econometric society, 153-161.
>
> [3] Nawata, K. (1994). Estimation of sample selection bias models by the maximum likelihood estimator and Heckman's two-step estimator. Economics Letters, 45(1), 33-40.

---

> > ### Author Response · Authors · 2022-11-18
> > **Thank you for the insightful comments. Here is the response to Reviewer CmDk (2/2).**
> >
> > > Q3) What does it mean by "non-random"? The proposed method treats all the variables as random variables. It sounds contradictory to me.
> >
> > A3) We use the term _non-random_ in the context of _non-random sample selection_, which means that the probability of any subject $i$ being selected from the population $\mathcal{P}\_{XY}$ to domain $k$ is not random. Denote $S^{k}$ as a binary random variable indicating whether a subject belongs to domain $k$. In a random sampling process, $P(S^{k}\_{i}=1)$ is independent and identically distributed (_iid_ or random selection). In the presence of non-random selection, the distribution of $(X,Y)$ in domain $k$ is a conditional distribution of $(X,Y)$ given $S^{k}=1$, which often does not equal to $\mathcal{P}\_{XY}$. Consequently, this leads to distributional shifts across domains: $\mathcal{P}\_{XY}^{j} \neq \mathcal{P}\_{XY}^{k}$ for $k \neq j$. More specifically, the proposed method can accommodate the scenarios that the non-random sampling probability $P(S^{k}\_{i}=1)$ is related to both $X$ and $Y$.
> >
> > > Q4) "A naively trained model, optimizing the prediction performance conditional on the selection mechanisms of the domains in the training data, will fail to perform similarly if sample selection of a testing domain is under a different selection mechanism.": I could not understand this sentence.
> >
> > A4) We agree that this sentence is unclear. What we meant is "An ERM model trained on $\mathcal{P}\_{XY}^{k}$ may fail to perform similarly if it is used on another domain with $\mathcal{P}\_{XY}^{j}$ when $\mathcal{P}\_{XY}^{k} \neq \mathcal{P}\_{XY}^{j}$. We have re-written the Introduction section to clarify.
> >
> > > Q5) "In such cases, we use $P\_{X}^{k}$ for $k=1,\dotsc,K$ as a surrogate, implying that we learn selection models by contrasting one domain to the remaining source domains.": This is not very clear to me. It would be nice if there were more explanations since, I suppose, this is part of the proposed method.
> >
> > A5) Please refer to response **A1** for details. We agree that this was unclear and have added Assumption 4 in the revised manuscript. Conceptually, Assumption 4 means that the same selection models $g_{k}$ can be learned by contrasting domain $k$ to the remaining domains in the training data ($K\setminus \{k\}$) as by contrasting domain $k$ to the remaining domains  in the population ($L\setminus \{k\}$) for $k=1 ,\dotsc, K$.

---

### Official Review · Reviewer_2hmL · 2022-10-24

**Confidence:** 4
**Correctness:** 3
**Technical Novelty And Significance:** 3
**Empirical Novelty And Significance:** 3
**Recommendation:** 8

**Clarity, Quality, Novelty And Reproducibility:**

Most of the parts are clear and well-written. No source code available and this approach is not easy to implement.

**Strength And Weaknesses:**

### Strength

1. This paper is well motivated. It is interesting to see that authors formulate this problem as a sample selection problem.
2. The theory part is novel, with solid proof.
3. Experiments show its effectiveness.

### Weakness

1. The computation complexity is not introduced in the paper. From what I see, there are two feature extractors in the method, while comparison methods only adopt one. So, what's the time complexity and number of parameters comparison? Are the experiments fair?
2. The latest comparison method is Fish, which is in 2021. There are other approaches more latest than this: SWAD and L2A-OT, for instance. Authors should cite and compare with them
3. There are no ablation studies to show the actual contribution of each component in the method.

**Summary Of The Paper:**

This paper casts domain generalization as a sample selection problem. It starts from problem formulation which is then disentangled into two terms: sample selection loss and joint function loss. Authors designed a dual-branch network to implement this idea. Experiments are done on WILDs benchmark, showing its effectiveness against its counterparts.

**Summary Of The Review:**

I think this paper is novel in formulating the DG problem as a sample selection one. It has its own theory and derivation, with working algorithms.

=== Comments after rebuttal: I decided to increase the score to 8 given that authors addressed my concerns well.

---

> ### Author Response · Authors · 2022-11-18
> **Thank you for the insightful comments. Here is the response to Reviewer 2hmL.**
>
> > Q1) The computation complexity is not introduced in the paper. From what I see, there are two feature extractors in the method, while comparison methods only adopt one. So, what's the time complexity and number of parameters comparison? Are the experiments fair?
>
> A1) Thank you for the comment.  Given $K$ training domains, the proposed method mainly consists of two components: 1) $K$ domain selection models $\mathbf{g}=\\{g\_{k}\\}_{k=1}^{K}$ and 2) an outcome prediction model $f$. More specifically, the domain selection models share a common feature extractor $\varphi\_{g}:\mathcal{X}\rightarrow\mathcal{Z}\_{g}$ such that $g\_{k}(\cdot)=\omega\_{k} \circ \varphi\_{g}(\cdot)$ for $k\in\\{1,\dotsc,K\\}$, which helps reduce both time and model complexity when $K$ is large. In other words, computational complexity of the proposed method does not scale with the number of training domains, since only a single forward pass $\varphi\_{g}(\boldsymbol{x})$ is necessary to compute all domain selection probabilities. The outcome prediction model has its own feature extractor $\varphi\_{f}:\mathcal{X}\rightarrow\mathcal{Z}\_{f}$ which is followed by a linear classifier $\omega\_{f}:\mathcal{Z}\_{f}\rightarrow\mathcal{Y}$. In general, $\varphi\_{g}$ and $\varphi\_{f}$ are allowed to have different neural architectures with different numbers of parameters. However, we found that simply using the same architecture (but learning different parameters) works well in practice (e.g., DenseNet-121 for Camelyon17). Therefore, in all our experiments, HeckmanDG has roughly twice as much trainable parameters as a method involving $f$ only. We would like to emphasize that $\mathbf{g}$ is only used during the training phase to guide $f$. At the testing stage, only $\hat{f}(\boldsymbol{x})$ is used to make predictions. In our empirical experiments, our $\hat{f}$ uses the same neural architectures as all competing methods, rendering all comparisons in the experiments fair. We have revised the manuscript to include this information in the second paragraph (Neural Network Architecture) of the Optimization section.
>
>
> > Q2) The latest comparison method is Fish, which is in 2021. There are other approaches more latest than this: SWAD and L2A-OT, for instance. Authors should cite and compare with them.
>
> A2) Thank you for informing us of more recent methods to compare with. We have added SWAD [1] and L2A-OT [2] to our list of comparative methods, whose results can be found in Tables 2, 3, and 4 in the Experiments section. Implementation details are stated in Appendix A.4. We did not run L2A-OT on the PovertMap dataset since it is only applicable to classification problems.
>
>
> > Q3) There are no ablation studies to show the actual contribution of each component in the method.
>
> A3) In Section 7, we conducted a series of ablation studies of our proposed method. We first showed the effectiveness of the two-step optimization procedure in Figure 3-(a) and 3-(b). We then demonstrated how the quality of the domain selection model $\mathbf{\hat{g}}$ impacts the performance of outcome prediction model $\hat{f}$, as show in Figure 3-(c), supporting our claim that estimating the selection probabilities correctly is important in improving performance of the outcome prediction model.
>
> [1] Cha, J., Chun, S., Lee, K., Cho, H. C., Park, S., Lee, Y., & Park, S. (2021). Swad: Domain generalization by seeking flat minima. Advances in Neural Information Processing Systems, 34, 22405-22418.
>
> [2] Zhou, K., Yang, Y., Hospedales, T., & Xiang, T. (2020, August). Learning to generate novel domains for domain generalization. In European conference on computer vision (pp. 561-578). Springer, Cham.

---

> > ### Comment · Reviewer_2hmL · 2022-11-18
> > **Response**
> >
> > Thank you for your feedback, which largely resolved my concerns. I would like to increase my score as long as the complexity analysis is included in the paper. Maybe it's even better with some empirical time complexity analysis?

---

> > > ### Author Response · Authors · 2022-11-19
> > > **Complexity Analysis**
> > >
> > > Thank you for your comment. We have added the complexity analysis in the revised manuscript. We are also working on the empirical time complexity analysis, but may not be able to finish it by the 1st discussion deadline (11/18). We will provide the results in an additional comment in a few days.

---

> > > > ### Author Response · Authors · 2022-12-02
> > > > **Complexity Analysis (Empirical)**
> > > >
> > > > Thank you for your comment. As shown in first table below, we observed that our proposed method requires approximately $1.4$ times longer per-batch training time compared to ERM with a single model $f$. Meanwhile, the running time on the testing data is comparable. In the second table, we show the observed per-batch computation time of $\mathbf{g}$ across the four datasets used in our work. For a fair comparison, the experiments were conducted with ResNet-50 feature extractors using a batch size of 32 and a unified input size of $224\times 224$. By using a shared feature extractor $\varphi_g$, we confirmed that the computation time does not scale with the number of training domains. We will add these tables to the revised manuscript Appendix.
> > > >
> > > > | Method | Training Time | Testing Time |
> > > > | :-------- | :--------------: | :-------------: |
> > > > | ERM                        | 0.094 ($\pm$0.018) | 0.015 ($\pm$0.009) |
> > > > | HeckmanDG (ours) | 0.131 ($\pm$0.017) | 0.015 ($\pm$0.008) |
> > > >
> > > > | Dataset                     | Camelyon17 | PovertyMap | RxRx1 | iWildCam |
> > > > | :-------:                      | :-------------: | :-------------: | :------: | :----------: |
> > > > | Training domains (K) | 3                    | 13                 | 33       | 243           |
> > > > | Time | 0.091 ($\pm$0.038) | 0.093 ($\pm$0.036) | 0.101 ($\pm$0.021) | 0.105 ($\pm$0.068) |

---

> > > > > ### Comment · Reviewer_2hmL · 2022-12-03
> > > > > **Thank you for your response**
> > > > >
> > > > > I would like to thank authors for the detailed complexity analysis. It seems that HeckmanDG shows comparable efficiency w.r.t. ERM. This is interesting. Please include this analysis in the future version.

---

### Official Review · Reviewer_Y91A · 2022-10-30

**Confidence:** 2
**Correctness:** 3
**Technical Novelty And Significance:** 4
**Empirical Novelty And Significance:** 3
**Recommendation:** 8

**Clarity, Quality, Novelty And Reproducibility:**

The paper is clearly written. The overall quality of paper is very good.
Based on my knowledge, the adoption of Heckman's methods is a novel idea.
I was not able to fully assess the reproducibility of the study, since the authors did not provide a code repository for testing.


**Strength And Weaknesses:**

The paper proposed a new method and did extensive experiments to demonstrate the effectiveness of the proposed method, which is a strength of the paper.

As the authors stated at the beginning of the paper, DG questions may be categorized into different categories. While the paper shows that the proposed method works in many cases, it is unclear to the reader, what exact scenario the proposed method may work and in which scenarios it will not. If the authors can shed some light on this direction, it will better navigate the readers about future applications of the method.

It will also be good if the authors can describe the pre-assumption of the methods more explicitly. The original Hackman's method has a number of assumptions, including the assumption of normal distribution etc. What are the implications for adoptions in the current scenarios? Hope the authors can add some discussions on such aspects.


**Summary Of The Paper:**

The authors consider domain generalization as a non-random sample selection problem. It is the first paper of kind to utilize this method for DG problem. Via assessment on both simulation and benchmarking data sets, the authors demonstrate the efficacy of our method both theoretically and empirically on simulated data and four challenging benchmarks.

**Summary Of The Review:**

The research work is well conducted and the paper is clearly written. Besides a couple of weakness, the research results are worth sharing with the broader audiences to raise discussions and potential further research in this direction.

---

> ### Author Response · Authors · 2022-11-18
> **Thank you for the insightful comments. Here is the response to Reviewer Y91A.**
>
> > Q1) As the authors stated at the beginning of the paper, DG questions may be categorized into different categories. While the paper shows that the proposed method works in many cases, it is unclear to the reader, what exact scenario the proposed method may work and in which scenarios it will not. If the authors can shed some light on this direction, it will better navigate the readers about future applications of the method.
>
> A1) We appreciate this suggestion, and have extended the descriptions on what scenarios the proposed method works, in the second paragraph of the Introduction.
> We consider domain generalization as a non-random sample selection problem. Let $\mathcal{P}\_{XY}$ represent the population data distribution. Non-random sample selection means that a sample (for any domain $k$) is not randomly selected from the population, i.e., the conditional data distribution of $(X,Y)$ given $S^{k}=1$, denoted as $\mathcal{P}^{k}\_{XY}$, does not equal to $\mathcal{P}\_{XY}$. Consequently, this leads to distributional shifts across domains: $\mathcal{P}\_{XY}^{j} \neq \mathcal{P}\_{XY}^{k}$ for $k \neq j$. Mathematically, distribution shifts across domains $\mathcal{P}\_{XY}^{k}$ may contain shifts in distributions of $X$ ($\mathcal{P}\_{X}^{k}$, covariate shift, [1]), and in the distributions of $Y$ conditional on $X$ ($\mathcal{P}\_{Y|X}^{k}$, [2]). We present graphical models in Figure 1 to conceptually illustrate the sources of distribution shifts, assuming the existence of latent factors confounding the relationship between $X$, $Y$, and domain ($S^{k}$). In Figure 1, $C\_1$ represents unobserved latent factors that correlate with $X$ and $S^{k}$, resulting in covariate shift. $C\_2$ correlates with $X$, $Y$, and $S^{k}$ simultaneously, entailing both covariate and concept shifts.
>
> The proposed Selection Guided Domain Generalization (SGDG) framework estimates a domain generalizable $f:\mathcal{X}\rightarrow\mathcal{Y}$ advocated by an optimal distributional shift estimate $\mathbf{g}$. By applying the Heckman correction framework, the resulting $f$ is robust to both sources of shifts. In contrast, other DG methods only emphasize one type of shift either by encouraging representations of features with similar distributions across domains (CORAL, DANN), or representations yielding similar classifiers across domains (IRM).
>
> Empirically, we confirmed that SGDG worked well in datasets where both shifts are possible (Figure 1). For example, in the Camelyon17 dataset, there may be variations in $X$ due to inconsistent acquisition processes such as staining differences ($C\_1$) across hospitals (covariate shift), and differences in other latent factors such as patient characteristics (age, gender, race, $C\_2$) that affect selection, covariate and the outcome distribution (covariate \& concept shift). Similarly, in the PovertMap dataset, latent factors such as the economic status of geographical locations ($C\_2$) could correlate with features, locations (domains), and outcome simultaneously. On the other hand, SGDG does not tackle cases where there may exist another latent factor ($C\_3$) correlating with selection $S$ and outcome $Y$ but not with $X$.
>
> > Q2) It will also be good if the authors can describe the pre-assumption of the methods more explicitly. The original Heckman's method has a number of assumptions, including the assumption of normal distribution etc. What are the implications for adoptions in the current scenarios? Hope the authors can add some discussions on such aspects.
>
> A2) We appreciate this suggestion and have explicitly added Assumptions 1-5 in the revised manuscript. In short, Theorem 1 (performance improvement of SGDG over SDRM) requires Assumptions 1 to 3.
> - **Assumption 1**: Mutually Exclusive Domain Membership: if $S^{k} = 1$, then $S^{j} = 0$ for all $j \neq k$ so that an instance can belong to one and only one domain.
> - **Assumption 2**: Independent Domain Sampling Processes: $S^{k} \perp S^{j}$ for all $j \neq k$.
> - **Assumption 3**: Decomposable Loss Function
>
> The parametric formula of HeckmanDG requires two additional assumptions:
> - **Assumption 4**: $\mathbf{g}$ learned from sets $1 \dots L$ is equal to $\mathbf{g}$ learned from sets $1 \dots K$.
> - **Assumption 5**: Joint distribution of $Y$ (or latent variables $\tilde{Y}$) and the selection latent variable $\tilde{S}$.
>
> [1] Bickel, S., Brückner, M., & Scheffer, T. (2009). Discriminative learning under covariate shift. Journal of Machine Learning Research, 10(9).
>
> [2] Moreno-Torres, J. G., Raeder, T., Alaiz-Rodríguez, R., Chawla, N. V., & Herrera, F. (2012). A unifying view on dataset shift in classification. Pattern recognition, 45(1), 521-530.

---

### Decision · Program_Chairs · 2023-01-20

**Decision:**

Accept: notable-top-25%

**Justification For Why Not Higher Score:**

Although the reviewers support this paper strongly, I feel that the assumptions it makes are too strong.

**Justification For Why Not Lower Score:**

Highest rated paper in my batch.

**Metareview: Summary, Strengths And Weaknesses:**

This paper proposes a DG method, termed HeckmanDG, that learns two models. The first model (g) predicts the domain membership of an input, and the second model (f) predicts the output of interest (Y).  The output of the first model is used when learning the second model to correct for domain selection bias.  The validity of the method hinges on some assumptions about the correlation between f and g (bottom of page 5 and A.2).  The effectiveness of the method has been empirically demonstrated, and a related theoretical result is also given.  All the reviewers are positive about this work.

**Note From Pc:**

if the above contains the word "oral" or "spotlight" please see: "oral" presentation means -> notable-top-5% and "spotlight" means -> notable-top-25%. As stated in our emails, we are disassociating presentation type from AC recommendations